# Establishing an Antimicrobial Stewardship Program in Sierra Leone: A Report of the Experience of a Low-Income Country in West Africa

**DOI:** 10.3390/antibiotics12030424

**Published:** 2023-02-21

**Authors:** Sulaiman Lakoh, Mohamed Bawoh, Hannah Lewis, Ishmael Jalloh, Catherine Thomas, Shuwary Barlatt, Abdulai Jalloh, Gibrilla F. Deen, James B. W. Russell, Mustapha S. Kabba, Moses N. P. Batema, Cecily Borgstein, Noah Sesay, Daniel Sesay, Navjeet K. Nagi, Emmanuel Firima, Suzanne Thomas

**Affiliations:** 1College of Medicine and Allied Health Sciences, University of Sierra Leone, Freetown, Sierra Leone; 2Ministry of Health and Sanitation, Government of Sierra Leone, Freetown, Sierra Leone; 3King’s Centre for Global Health and Health Partnerships, School of Life Course & Population Sciences, King’s College London, London SE1 1UL, UK; 4Pharmacy Department, King’s College Hospital NHS Foundation Trust, London SE5 9RS, UK; 5Division Clinical Epidemiology, University Hospital Basel, CH-4051 Basel, Switzerland; 6Faculty of Medicine, University of Basel, CH-4001 Basel, Switzerland; 7Clinical Research Unit, Department of Medicine, Swiss Tropical and Public Health Institute, CH-4051 Basel, Switzerland; 8Centre for Multidisciplinary Research and Innovation, Abuja 9000211, Nigeria

**Keywords:** global point prevalence survey, GPPS, antimicrobial stewardship program, AMS program, AMS team, Freetown, Sierra Leone

## Abstract

Antimicrobial Resistance (AMR) is a growing global health challenge that threatens to undo gains in human and animal health. Prevention and control of AMR requires functional antimicrobial stewardship (AMS) program, which is complex and often difficult to implement in low- and middle-income countries. We aimed to describe the processes of establishing and implementing an AMS program at Connaught Hospital in Sierra Leone. The project involved the setting up of an AMS program, capacity building and performing a global point prevalence survey (GPPS) at Sierra Leone’s national referral hospital. Connaught Hospital established a multidisciplinary AMS subcommittee in 2021 to provide AMS services such as awareness campaigns, education and training and review of guidelines. We performed a GPPS on 175 patients, of whom more than half (98, 56.0%) were prescribed an antibiotic: 63 (69.2%) in the surgical wards and 53 (51.2%) in the medical wards. Ceftriaxone (60, 34.3%) and metronidazole (53, 30.3%) were the most common antibiotics prescribed to patients. In conclusion, it is feasible to establish and implement an AMS program in low-income countries, where most hospitalized patients were prescribed an antibiotic.

## 1. Introduction

Antimicrobial Resistance (AMR) is a growing global health challenge that threatens to undo gains in human and animal health [1,2]. A recent global estimate reports that 4.95 million deaths annually are associated with AMR, of which 1.27 million are directly attributable to bacterial AMR [3]. By 2050, an estimated 10 million AMR-related deaths will occur each year, many in Africa and Asia, if action is not taken to address the problem [4].

To stem the growing burden of AMR, the World Health Organization (WHO) developed a global action plan in 2015 to support the AMR prevention and control efforts of its member states [2]. Among the five strategic principles of the global action plan, WHO recommends rational use of antimicrobials to prevent the development of resistance [2]. Antimicrobial Stewardship (AMS) is a major tool that combines interventions designed to ensure the appropriate use of antimicrobials to contain the emergence of resistance and maintain the efficacy of existing antimicrobials [4]. Establishing a functional AMS program is essential to reduce unnecessary antimicrobial use and thus the burden of AMR [5,6,7]. Despite the need for AMS, its implementation in low- and middle-income countries (LMICs) is complex and often difficult to achieve because AMS implementation is influenced by the local context of the weak health systems in these countries [5,6,7].

Sierra Leone, a low-income country in West Africa, is struggling to rebuild its health system following the adversity of the 2014–2016 Ebola outbreak and a decade-long civil war from 1991 to 2002 [8,9,10]. In 2018, Sierra Leone developed a National Strategic Plan to combat AMR, which is aligned with the WHO Global Action Plan [2,11]. Yet, there are no functional stewardship activities in any of the hospitals in Sierra Leone.

This situation, and the fact that our previous work reported many challenges with rational antibiotic prescribing prompted us to set up the only existing AMS program at Sierra Leone’s national referral hospital [12,13]. In this article, we aim to describe the processes of establishing and operationalizing an AMS program in the national referral hospital of Sierra Leone.

## 2. Results

### 2.1. Setting up and Operationalizing an AMS Program in a Low-Income Country

Connaught Hospital established an AMS subcommittee in 2021 to provide input to the Infection Prevention and Control (IPC), Water Sanitation and Hygiene (WASH) and AMS committee (this broader committee existed before but did not include AMS). Members of the subcommittee come from across the hospital, including nursing, medicine, pharmacy, laboratories, surgery and IPC. The subcommittee nominated an AMS lead and deputy. The terms of reference were agreed and the subcommittee aimed to meet every month. The hospital’s IPC and WASH committee incorporated AMS into its remit, and agreed that the lead for the AMS subcommittee would reports to the broader IPC, WASH and AMS committee on a quarterly basis, which subsequently feeds back to the senior management of the hospital.

The AMS Subcommittee supported the conduct of the GPPS and used the results as the basis for an AMS action plan. This subcommittee provides direction and oversight for the promotion and delivery of successful AMS services throughout the institution.

The subcommittee supports Connaught Hospital in identifying and implementing any national guidance on AMS, including identifying any resources needed to overcome constraints. The subcommittee also led hospital awareness campaigns and partnered with the nursing education team to launch the use of an AMS board game to raise awareness and educate nursing staff. In addition, the AMS subcommittee led the planning of educational events related to AMS, drawing on experts to speak, designing messaging and inviting stakeholders.

### 2.2. Designing AMS Related Policies and Guidelines

The IPC, WASH, and AMS broader committee developed terms of reference for the AMS subcommittee to support and define its operations. The need to review the surgical prophylaxis guidelines was identified through previous research at the hospital and the recent GPPS results [12,13]. It was decided to bring in various stakeholders for the review of the surgical prophylaxis guidelines. Stakeholders included representatives from Connaught Hospital, King’s Global Health Partnerships (KGHP), the WHO Country Office and Ministry of Health and Sanitation. The review of surgical prophylaxis guidelines for Connaught Hospital was conducted with the view to creating a national guideline which can be shared and implemented across all government hospitals/facilities offering surgical services in Sierra Leone. The involvement of key stakeholders enabled the validation and promotion of these guidelines at national level. Given that most of the relevant expertise for AMR is based in the capital of Sierra Leone, creation of a guideline that is applicable to other facilities, is evidence of a commitment to spread the benefits from this AMS program across the country.

We developed a five-year AMS action plan that is based on five strategic goals, including leadership commitment, AMS actions, education and training, monitoring and surveillance, and reporting feedback in healthcare settings.

### 2.3. Partnership and Funding

Connaught Hospital has an established long-term partnership with an international non-governmental organization, KGHP. Connaught Hospital, in collaboration with KGHP, the Young Pharmacists Group and the Pharmaceutical Society of Sierra Leone, secured funding from the Commonwealth Partnerships for Antimicrobial Stewardship. These partnerships are funded by the Fleming Fund (a UK aid program) and managed by the Tropical Health and Education Trust (THET) and the Commonwealth Pharmacists Association (CPA). This grant supported the work described. The hospital also leverages the activities of existing partnerships within the Ministry of Health and Sanitation and international health agencies such as the WHO.

### 2.4. AMS Champion Training

During the program, ten pharmacists (5 women; 5 men) were recruited and trained as AMS Champions, using Connaught Hospital as a training site.

Those recruited to the training program work in the pharmacy sector, including hospitals, community pharmacies, and administrative departments. The training program design was based on the WHO competency framework and curricula for healthcare workers and accredited by the Pharmaceutical Society of Sierra Leone [14]. This blended learning program included a range of learning activities: ward visits; group discussions; individual discussions with tutors about patient cases/AMS; case presentations and discussions; preparing, collecting data for, and analyzing the GPPS; community awareness raising activities; online continuous professional development (CPD) modules; and the AMS board game.

The AMS Champions together with the AMS Lead and Deputy led the data collection for the GPPS and supported the subcommittee in raising awareness about AMS and disseminating the GPPS findings in the hospital.

### 2.5. Global Point Prevalence Survey (GPPS)

During the course of the project, a GPPS was performed on 175 inpatients in the hospital. Their mean (SD) age was 33 ± 19 years. The majority were males (124, 70.9%), admitted to the surgical wards of the hospital (112, 64.0%) and aged 20–40 years (79, 45.1%).

More than half (98, 56.0%) of the patients were prescribed an antibiotic, most in the surgical wards (63, 69.2%). Antibiotics prescribed to patients in this study were mainly in the Access (85, 48.6%) or Watch (83, 47.4%) groups. The most common reason for prescribing antibiotics was to treat community-acquired infections (92, 52.5%) (Table 1). Ceftriaxone J01DD04 (60, 34.3%) and metronidazole P01AB01 (53, 30.3%) were the most common antibiotics prescribed to patients in this study (Table 2). Twenty-five (14.2%) cases did not specify the indications for antibiotic prescription. Only 25 (15.3%) patients were prescribed antibiotics according to hospital treatment guidelines.

In multivariable regression analysis, women were less likely to have an indication for prescribing antibiotics: [aOR 0.28, 95% CI (0.09–0.83); *p* = 0.022]. Indications for prescribing were less likely to be stated for the unclassified antimicrobial agents than Access antibiotics (Table 3).

## 3. Discussion

We report our experience in establishing and operationalizing an AMS program in Sierra Leone to add to the evidence base in the global literature. The implementation and operationalization of an AMS program in hospitals of low-income countries is important because AMS activities are critical to the achievement of the United Nations Sustainable Development Goals (Targets 3.1 to 3.3 and 3.8, and Goal 6), especially those focused on reducing AMR [15]. This report of the activities, for the only existing hospital AMS program in Sierra Leone at the moment, showed that it is feasible to establish and implement an AMS program in low-income countries when there is support from the local human resources. An AMS program was previously set up in a rural hospital in Sierra Leone but it ceased to exist after the departure of the foreign experts who carried out its activities, demonstrating that a sustainable AMS program is the one that receives support from the local hospital structures [16].

Research from hospitals across Sierra Leone, including Connaught Hospital, found deep-rooted challenges not only in antimicrobial prescribing and antibiotic resistance, but also in IPC [17,18,19,20,21,22]. This reinforces the fact that an AMS program should constitute a broad-based membership that includes IPC practitioners [23]. Furthermore, we reflect on the call in 2022 by the WHO Director-General to prioritize IPC as a key to health system strengthening and universal health coverage instrument and the idea that ‘every infection prevented is an antibiotic resistance avoided’ [24]. Consequently, we adopted a multidisciplinary approach in the selection of the AMS subcommittee members and in the implementation of the AMS activities, similar to the recommendations made by Kirchhelle et al. [25]. Our AMS program is unique in that it operates within the remit of a broader IPC, WASH and AMS committee that interweaves IPC, WASH and AMS activities within an integrated framework at the hospital. Through this broader committee, the three subcommittees can share experiences, gain a better understanding of the operations of the different subcommittees, and generally improve the outcomes of their interventions.

We developed policies, updated guidelines and trained healthcare workers to support the implementation of our AMS program. In his guide on how to start a hospital AMS program, Mendelson recommends training and resource mobilization as part of the preparation for implementing AMS activities [23].

Our AMS program conducted a GPPS, which was critical in the external validation of our AMS experience, because it was easy to complete and the data collection software automatically generated a report which enabled comparison of antimicrobial prescribing at Connaught Hospital with other hospitals in the African continent and worldwide. In this hospital-wide GPPS, 56% of hospitalized patients were prescribed antibiotics, compared with 82% reported in 2017 [12]. The difference in the prevalence between the two studies could be attributed to the differences in the methods applied (point prevalence vs. period prevalence) and ongoing AMS activities or perhaps it could be due to the availability of more medical specialists to oversee antibiotic prescribing. Likewise, the departure of low-level medical staff known as community health officers (CHOs) who historically prescribed large quantities of antibiotics in the emergency department of Connaught Hospital could explain the difference in prevalence between the two studies. Yet, this prevalence of 56% is still higher than the African average of 42% [26]. In addition, antibiotic use remained higher among inpatients in adult surgical wards (69.6%) compared with adult medical wards (51.9%). This discrepancy may be related to the surgical team’s concern about poor IPC and environmental conditions in the wards or operating theatres leading to unnecessary and prolonged use of prophylactic antibiotics. We join the call to reshape the global health architecture to reduce reliance on the use of antibiotics for patient care and encourage non-medicinal care, such as providing supports to patients and addressing their concerns [27]. Combined with the results of a previous study, this point prevalence survey showed an ongoing high rate of antibiotic use among surgical patients and provided the stimulus for reviewing and updating guidelines for antibiotic prophylaxis in surgery [13]. We believe that the inappropriate use of antibiotics in the surgical wards will reduce with the implementation of targeted AMS activities.

Although no Reserve antibiotics were prescribed in this study, which may reflect unavailability, the prescription of non-antibacterial agents such artesunate and acyclovir without documented indication calls for expanded stewardship activities.

Our AMS program is not without challenges or limitations. One of the main challenges in the implementation of the AMS program is the unavailability of financial support from local sources as the Government of Sierra Leone does not provide specific budgetary support for AMS activities. As shown by GPPS, less than 11% of antibiotic prescriptions were supported by microbiology or other laboratory tests, showing the hospital’s limited laboratory capacity to support AMS activities. Owing to the funder’s requirements, focused primarily on pharmacists, we could not directly fulfil the multidisciplinary principles of AMS in our AMS champion training program. Nonetheless, the AMS champions provided support for certain elements such as the GPPS and spent time on the wards talking to doctors and nurses about the prescribing issues they identified. Finally, the limited availability of the AMS subcommittee staff has at times affected the conduct of the AMS program at Connaught Hospital.

## 4. Methods and Materials

### 4.1. Project Design and Settings

The project involved the setting up of an AMS program, capacity building and performing a point prevalence survey at the Sierra Leone’s national referral hospital for medical and surgical cases (Figure 1). Sierra Leone is divided into five geographical regions, of which the Western Area is the most densely populated with a population of approximately 1.5 million [28,29]. The public health system is divided into primary, secondary and tertiary levels of care.

The project was implemented in Connaught Hospital, which is Sierra Leone’s national referral hospital in the Western Area of Sierra Leone with a capacity of 300 beds. Connaught Hospital is affiliated with the College of Medicine and Allied Health Sciences of the University of Sierra Leone. It has 12 wards, an intensive care unit, a medical observation department, and an accident and emergency department [30].

### 4.2. Global Point Prevalence Survey (GPPS) Data Collection and Analysis

We conducted a point prevalence survey using an online data collection tool provided by the GPPS central platform [https://www.global-pps.com/ (accessed on 20 January 2023)] to determine antibiotic prescribing patterns in the wards of Connaught Hospital on 11 March 2022. Indications for antibiotic prescription were divided into community-acquired infections, healthcare-associated infections, and medical or surgical prophylaxis. Cases for which no antibiotic prescription indication was specified were designated as unknown cases.

Upon completion of the data collection, the data was exported into a Microsoft Excel file (Microsoft, Redmond, WA, USA), cleaned, coded and then transferred into Stata Version 16 (StataCorp LLC, College Station, TX, USA) for analysis. Descriptive statistics such as mean and standard deviation were used for normal continuous variables; median and interquartile ranges were used for nonparametric variables; and frequencies and percentages were used for categorical variables. The prescribed antibiotics were categorized using the WHO’s Access, Watch and Reserve (AWaRe) framework and the Anatomic Therapeutic and Chemical Classification (ATC) System as described in the WHO’s Essential Model List [31,32].

A multivariable logistic regression model was built to predict factors associated with stated indications for antibiotic prescription. All variables were sequentially entered into the model, and if the overall model fit was improved according to the Akaike Information Criterion (AIC), all variables were retained in the final model. All tests were two-tailed and significance was set at *p* < 0.05.

## 5. Conclusions and Recommendations

In conclusion, it is feasible to establish and implement an AMS program in a low-income country such as Sierra Leone, where hospitalized patients are likely to be given antibiotics.

We recommend the following in establishing an AMS program in a low-income country:
(a)Use existing structures, such as IPC committee and teams (where available).(b)Establish partnership with local and international agencies.(c)Secure funding.(d)Establish the policy environment such as terms of reference for the AMS team.(e)Establish a multidisciplinary AMS team.(f)Train healthcare workers.(g)Create an action plan to guide activities.(h)Use pre-existing GPPS methodologies to track progress of the AMS work.

## Figures and Tables

**Figure 1 antibiotics-12-00424-f001:**
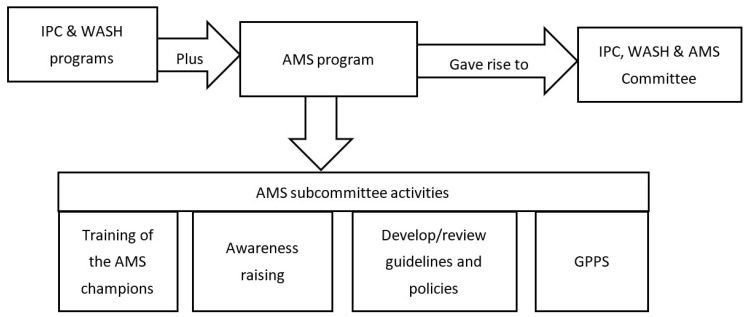
Flow diagram showing AMS subcommittee activities and related structures. AMS = Antimicrobial stewardship; GPPS = Global point prevalence survey; IPC = Infection Prevention and Control; WASH = Water Hygiene and Sanitation.

**Table 1 antibiotics-12-00424-t001:** Demographic characteristics of participants and antibiotics prescribing patterns.

Parameter	Total	MW	SW	ICU
	*n*(%)	*n* (%)	*n* (%)	*n*(%)
Overall total	175 (100)	60 (34.3)	112 (64.0)	3 (1.7)
Sex				
Female	51 (29.1)	30 (50.0)	18 (16.1)	3 (100.0)
Male	124 (70.9)	30 (50.0)	94 (83.9)	0 (0.0)
Age				
<20	45 (25.7)	7 (11.7)	38 (33.9)	0 (0.0)
20–40	79 (45.2)	29 (48.3)	49 (43.8)	1 (33.3)
41–60	27 (15.4)	14 (23.3)	13 (11.6)	0 (0.0)
>60	24 (13.7)	10 (16.7)	12 (10.7)	2 (66.7)
Mean (SD)	33 (18.0)	-	-	-
Prescriptions based on laboratory results				
Yes	19 (10.9)	5 (8.3)	14 (12.5)	0 (0.0)
No	156 (89.1)	55 (91.7)	98 (87.5)	3 (100)
Indications for prescribing antibiotics				
CAI	92 (52.5)	37 (61.7)	52 (46.4)	0 (0.0)
HAI	30 (17.1)	16 (26.7)	22 (19.6)	3 (100.0)
Medical prophylaxis	1 (0.6)	1 (1.7)	0 (0.0)	0 (0.0)
Surgical prophylaxis	28 (16.0)	0 (0.0)	28 (25.0)	0 (0.0)
Unknown	24 (13.7)	8 (13.3)	16 (14.3)	0 (0.0)
Documented antibiotic stop/review date				
Yes	38 (21.7)	16 (26.7)	22 (19.6)	0 (0.0)
No	137 (78.3)	44 (73.3)	90 (80.4)	3 (100)
Reason in notes				
Yes	119 (68.0)	40 (66.7)	76 (67.9)	3 (100)
No	56 (32.0)	20 (33.3)	36 (32.1)	0 (0.0)
AWaRe				
Access	85 (48.6)	26 (43.3)	58 (51.8)	1 (33.3)
Watch	83 (47.4)	29 (48.3)	52 (46.4)	2 (66.7)
Not classified	7 (4.0)	5 (8.3)	2 (1.8)	0 (0.0)
Prevalence of antibiotic use	98 (56.0)	33 (51.6)	63 (69.2)	2 (100.0)
Dose per day (IQR)	2 (2–3)	2 (2–3)	2 (2–3)	2.5 (2–3)
Route				
Parenteral	130 (73.9)	38 (63.3)	88 (78.6)	4 (100)
Oral	46 (26.1)	22 (36.7)	24 (21.4)	0 (0)

MW = Medical ward; SW =Surgical ward; ICU = Intensive Care Unit. CAI = Community acquired infections; HAI =Healthcare Infections; IQR = Interquartile range.

**Table 2 antibiotics-12-00424-t002:** Frequency of administration of antibiotics and AWaRe category.

Variable	ATC Code	AWaRe Category	Drug Administration			
			Total *n* (%)	MW *n* (%)	SW *n* (%)	ICU *N* (%)
Acyclovir	-	-	1 (0.6)	1 (1.7)	-	-
Amoxicillin	J01CA04	Access	18 (10.3)	12 (20.0)	6 (5.4)	-
Amoxicillin-clavulanate	J01CR02	Access	8 (4.6)	3 (5.0)	5 (4.5)	-
Ampicillin	J01CA01	Access	1 (0.6)	-	1 (0.9)	-
Ampicillin-cloxacillin	J01CR50	Access	2 (1.1)	-	2 (1.8)	-
Artemether-lumefantrine	-	-	2 (1.1)	2 (3.3)	-	-
Artesunate	-	-	1 (0.6)	1 (1.7)	-	-
Azithromycin	J01DB01	Watch	8 (4.6)	8 (13.8)	-	-
Cefepime	J01DE01	Watch	1 (0.6)	-	1 (0.9)	-
Cefoxitin	J01DC01	Watch	1 (0.6)	1 (1.7)	-	-
Cefixime	J01DD08	Watch	1 (0.6)	-	1 (0.9)	-
Ceftriaxone	J01DD04	Watch	60 (34.3)	15 (25.0)	43 (38.4)	2 (66.7)
Clarithromycin	J01FA09	Watch	12 (1.2)	1 (1.7)	1 (0.9)	-
Erythromycin	J01FA01	Watch	2 (1.1)	-	2 (1.8)	-
Flucloxacillin	J01CF05	Access	2 (1.1)	2 (3.3)	-	-
Fluconazole	-	-	2 (1.1)	1 (1.7)	-	-
Gentamycin	J01GB03	Access	2 (1.1)	-	2 (1.8)	-
Levofloxacin	J01MA12	Watch	4 (2.3)	2 (3.3)	2 (1.8)	-
Metronidazole	P01AB01	Access	53 (30.3)	9(15.5)	43 (38.4)	1 (33.3)
Ofloxacin	J01MA01	Watch	1 (0.6)	-	1 (1.7)	-
Piperacillin-tazobactam	J01CR05	Watch	1 (0.6)	-	1 (0.9)	
TDF+3TC+DTG	-	Watch	1 (0.6)	1 (1.7)	-	-
Tinidazole	P01AB02	Access	22 (2.2)	22 (7.3)	1 (0.9)	-
Tinifloxacin	J01RA13	Access	2 (0.2)	2 (0.7)	-	-
Vancomycin	J01XA01	Watch	1 (0.6)	-	1 (0.9)	

MW = Medical ward; SW =Surgical ward; ICU = Intensive Care Unit; TDF + 3TC + DTG = Tenofovir + Lamivudine + Dolutegravir; ATC = Anatomic Therapeutic and Chemical classification of drugs.

**Table 3 antibiotics-12-00424-t003:** Multivariable regression of presence of prescription indication.

Parameter	Indication for Antibiotic Prescription		aOR (95% CI)	*p*
	Yes (%)	No (%)		
Age *	32 (20)	38 (16)	0.98 (0.96–1.01)	0.221
Sex				
Male	111 (73.5)	13 (52)	1	-
Female	40 (26.5)	12 (48)	0.28 (0.09–0.83)	0.022
Ward				
Medical	52 (34.4)	8 (32)	1	-
Surgical	96 (63.6)	16 (64)	0.47 (0.15–1.45)	0.188
ICU	3 (2.0)	1 (4)	0.76 (0.06–9.47)	0.829
Reason in notes				
No	43 (28.5)	14 (56)	1	-
Yes	108 (71.5)	11 (44)	3.78 (1.47–9.77)	0.006
Prescription based on lab. results				
No	137 (90.7)	20 (80)		
Yes	14 (9.3)	5 (20)	0.25 (0.07–0.91)	0.035
AWaRe				
Access	73 (44.3)	12 (48)	1	
Watch	74 (49.0)	9 (36)	1.25 (0.46–3.38)	0.658
Not classified	4 (2.7)	4 (16)	0.16 (0.03–0.84)	0.031

* Reported as mean (standard deviation); ICU = intensive care unit.

## Data Availability

Data is available at the University of Sierra Leone repository and will be available on request.

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
