# Peer review of "Establishing an Antimicrobial Stewardship Program in Sierra Leone: A Report of the Experience of a Low-Income Country in West Africa"

_antibiotics, 2023, doi:10.3390/antibiotics12030424_

Round 1

Reviewer 1 Report

Revisions

I think that this article has a big importance and interest, since, as we know, antimicrobial resistance is a great challenge for healthcare workers, not only in high income countries, but also in low- and medium-income countries, where resources are limited. Demonstrating that the establishment of antimicrobial stewardship programs is possible even in these countries, represents a huge moral slap even for high-income countries, where these measures have not yet been adopted.

Nevertheless, I suggest some minor revisions:

Line 71. Please make sure that all abbreviations have been coded (i.e. IPC)

Line 150. Please edit the figure, because some sentences are unreadable.

Table 1. Please check numbers and percentages.

Author Response

 Comment: I think that this article has a big importance and interest, since, as we know, antimicrobial resistance is a great challenge for healthcare workers, not only in high income countries, but also in low- and medium-income countries, where resources are limited. Demonstrating that the establishment of antimicrobial stewardship programs is possible even in these countries, represents a huge moral slap even for high-income countries, where these measures have not yet been adopted.

·      Response: Thank you

·      Comment: Line 71. Please make sure that all abbreviations have been coded (i.e. IPC)

·      Response: We have addressed this issue in the manuscript.

·      Comments: Line 150. Please edit the figure, because some sentences are unreadable.

·      Response: Thank you for this observation. This has been taken into consideration.

·      Comments: Table 1. Please check numbers and percentages.

Response: We have corrected this, thank you.

Reviewer 2 Report

Thank you for the opportunity to review this paper. The authors of this manuscript “set up the only existing AMS program at Sierra Leone’s national referral hospital” (p2, lines 64-5). The accomplishments are clear, and the success of the endeavor is enviable given the constraints. While this article is an important example of the value of implementation of AMS and acknowledges the “complex” and “difficult” (p1, line 27) aspects of AMS implementation, the article does not put this accomplishment in a broader, global context. Additionally, there is little specificity in terms of describing the implementation of this AMS program, thus making it difficult to assess the success of the enterprise. In its current state, the article is more an announcement than it is a research article.

Below are further comments and suggestions for revision:

1)     The authors acknowledge involvement in the creation of the program, but it is not clear how the conduct of the research for this article is related to the running of the program. Funders are not stated by name. The discussion of conversations with the AMS committee suggests that the authors are influential in the AMS program but there is not much detail there.

2)     The global point prevalence survey is an example of measuring antibiotic use, and yet in the title the authors claim to measure experience. How did the authors measure experience? Where is that data?

3)     The order of sections is somewhat unconventional, with materials and methods coming after results and discussion.

4)     The article ends with a generic list of how to implement programs in a low-income country. I suggest adding citations by others who have implemented stewardship programs and/or directed antibiotic use policy in low-income countries. Start by looking at:

*Setting the standard: multidisciplinary hallmarks for structural, equitable and tracked antibiotic policy (Kirchelle et al)

*And the authors: Justin Dixon, Christine Nabirye, Eleanor MacPherson, Komatra Chuengsatiansup, etc.

Author Response

•    Comments: Thank you for the opportunity to review this paper. The authors of this manuscript “set up the only existing AMS program at Sierra Leone’s national referral hospital” (p2, lines 64-5). The accomplishments are clear, and the success of the endeavor is enviable given the constraints. While this article is an important example of the value of implementation of AMS and acknowledges the “complex” and “difficult” (p1, line 27) aspects of AMS implementation, the article does not put this accomplishment in a broader, global context. Additionally, there is little specificity in terms of describing the implementation of this AMS program, thus making it difficult to assess the success of the enterprise. In its current state, the article is more an announcement than it is a research article.
•    Response: Thank you so much for your comment. We have added some information to reshape the global health architecture to shift the pressure from antibiotic use to other forms of patient care (Lines 221 to 228).  We use the GPPS because it is easy to complete, the data collection tool is set up already, the results are processed by a central body and report is automatically generated enabling comparison to other hospitals on the African continent and worldwide. We have added this information to the manuscript (Lines 204 to 207). Furthermore, throughout the manuscript, we emphasized the global context of AMR. In the introduction section, we provide information on the global literature on lines 42 to 49 and line 62 to 64. We used the WHO competency framework to train the AMS champions (Line 120 to 122). We cited global data in the discussion section (lines 174 to 178) and reflect on the call in 2022 by the WHO Director-General to prioritize IPC as a key health system strengthening and universal health coverage instrument and the idea that ‘every infection prevented is an antibiotic resistance avoided’ (Lines 189 to 193).  We feel that all of this information put our accomplishment on the global context. 
•    Comments: The authors acknowledge involvement in the creation of the program, but it is not clear how the conduct of the research for this article is related to the running of the program. Funders are not stated by name. The discussion of conversations with the AMS committee suggests that the authors are influential in the AMS program but there is not much detail there.
•    Response: Although we are not too clear what message you are trying to convey here, we generated this manuscript by putting together activities of the AMS sub-committee at Connaught Hospital. Not all the authors of this paper are members of the AMS sub-committee. We felt we did not name any funder by name. Details about funders is included in section 2.3 and under the funding section in the footnotes of the paper. 
•    Comments: The global point prevalence survey is an example of measuring antibiotic use, and yet in the title the authors claim to measure experience. How did the authors measure experience? Where is that data? 
•    Response: We conducted a point prevalence survey and compared the results of this survey with previous evidence from Sierra Leone and other countries. We thought it was important to let the audience know that using pre-existing GPPS methods can help track the progress of AMS program. Throughout the manuscript, we highlight the activities we undertake to set up an AMS in a low-income setting. Additionally, we highlighted our AMS activities, including AMS champion training, policy document development, awareness raising, and more. We feel that this information is worth sharing as lessons learned from an AMS program in low-income countries. 
•    Comments: The order of sections is somewhat unconventional, with materials and methods coming after results and discussion.
•    Response: We followed the format provided by the journal, but we can reformat the manuscript into any format recommended by the journal’s editorial team. 
•    Comments: The article ends with a generic list of how to implement programs in a low-income country. I suggest adding citations by others who have implemented stewardship programs and/or directed antibiotic use policy in low-income countries. Start by looking at:
*Setting the standard: multidisciplinary hallmarks for structural, equitable and tracked antibiotic policy (Kirchelle et al)
*And the authors: Justin Dixon, Christine Nabirye, Eleanor MacPherson, Komatra Chuengsatiansup, etc.
•    Response: Thank you for this comment. We have added details on this in the manuscript (Lines 239 to 245, Page 6 and added references 25 and 27)

Round 2

Reviewer 2 Report

Thank you for the opportunity to review this paper. I still believe this article manuscript to be a valuable contribution to the literature on antimicrobial stewardship and antibiotic use. The changes made in terms of adding select citations and few clarifying sentences do contribute to contextualizing the research. Additional clarity could be added in terms of whose "experience" is referenced in the title and the specific methodological procedures that were followed in what months. With these changes the manuscript could be better understood by readers.

Note: Kirchelle et al are multiple authors, the edit in the main body of the article manuscript only states "Kirchelle."